

# Studying the large-scale effect of leaf thermoregulation using an Earth system model

Marvin Heidkamp[1], Felix Ament[1,2], Philipp de Vrese[1], and Andreas Chlond[1]

[1]Max Planck Institute for Meteorology, Hamburg, Germany
[2]Meteorological Institute, CEN, University of Hamburg, Germany

**Correspondence:** Marvin Heidkamp (marvin.heidkamp@mpimet.mpg.de)

**Abstract.** Plants have the ability to regulate heat and water losses. This process also known as "leaf thermoregulation" helps to maintain the leaf temperature within an optimal range. In a number of laboratory and field experiments, the leaf temperature has been found to deviate substantially from the ambient temperature. In the present study, we address the question of whether the negative correlation between the leaf temperature excess and the ambient air temperature, which is characteristic of leaf

thermoregulation, constitutes a robust feature at larger scales, across a broad range of atmospheric conditions and canopy characteristics. To this end, we developed a new dual-source canopy layer energy balance scheme (CEBa) and implemented it into JSBACH, the land component of the Max Planck Institute for Meteorology's Earth system model (MPI-ESM). The approach calculates the temperature and humidity in the ambient canopy air space, the temperature of the ground surface, and the temperature of the leaf as well as the energy and moisture fluxes between the different compartments. Here leaf

thermoregulation is investigated using different modeling approaches, namely a zero-dimensional instantaneous solution of the energy balance as well as offline FLUXNET site experiments and coupled global simulations. With the help of the simulations at the site-level, we can show that the model is capable of reproducing the effect of leaf thermoregulation even though the simulated signal at the canopy scale is less pronounced than indicated by measurements at the leaf scale. However, on a global scale and over longer-timescales, this negative correlation is only simulated in idealized setups that neglect limitations on the

plant available water, and even then, the signal is less pronounced than indicated by the short-term observations of individual leaves. When accounting for moisture limitations, we predominantly find positive correlations between leaf temperature excess and the ambient air temperature.

## 1   Introduction

The canopy is commonly defined as "*the community of aboveground plant organs*" (Campbell et al., 1989) with over two-thirds of the terrestrial surface being covered by vegetation (Huete et al., 2004). The canopy influences the turbulent exchange of energy, water, and momentum between the land surface and the overlying air mass via a number of biogeophysical and




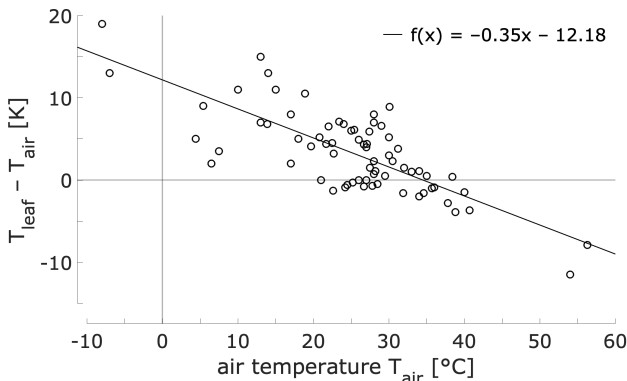

**Figure 1. Relation between leaf temperature excess and air temperature**, based on short-term measurements of isolated and water-unstressed sunlit plants. Figure redrawn from Linacre (1967).

biogeochemical processes. Furthermore, its complex structure and the resultant shading strongly affect the terrestrial radiation budget, largely regulating the distribution of incoming solar energy into ground and turbulent heat fluxes. This gives the canopy

a key role in controlling surface and below-ground temperatures, which in turn affect the diurnal variation of the boundary layer development and important atmospheric processes such as cloud formation and convection.

     Moreover, the canopy determines key characteristics of the hydrological cycle, via a strong influence on evaporation and transpiration (e.g., Oki and Kanae, 2006). This affects dynamics and thermodynamics of the climate system (Chahine, 1992), especially because soil moisture is part of several feedbacks on the local, regional, and even on the global scale (Seneviratne

et al., 2010; Lawrence et al., 2007). The canopy also constitutes an important element of the global carbon cycle, as substantial amounts of atmospheric carbon are taken up by the leaves within the canopy layer. Woody plants alone assimilate about $120\,\mathrm{Gt}$ carbon per year through the process of photosynthesis (e.g., Schimel et al., 2001), with net primary productivity (NPP) being largely constrained by atmospheric $CO_2$ concentrations and the available soil moisture, but also by the prevailing leaf temperatures (e.g., Farquhar et al., 1980; Farquhar and Sharkey, 1982; Ball et al., 1987).

The first detailed studies on the energy balance of leaves and the heat transfer between the leaf and its environment date back to the 1960s. Here Gates (1962) was amongst the first to show that leaf temperatures can deviate substantially from the temperature of the surrounding air: Sunlit leaves were up to 20 °C warmer than the ambient air, while shaded leaves were found to be on average 1.5 °C colder than the air temperature. These temperature differences occur on time scales ranging from seconds to minutes and can vary significantly throughout the day due to varying environmental conditions such as solar and

thermal radiation, air temperature, wind speed, and water vapor content (Gates, 1965). Linacre (1964) introduced the concept of "leaf thermoregulation" (LT), demonstrating a distinct negative correlation between the ambient air temperature and the leaf temperature excess – which is the difference between leaf and ambient air temperature – for water-unstressed, photosynthetic plants (Fig. 1). This study also showed that the leaf temperature excess changes its sign, allowing the leaf to be warmer than the surrounding air in cold environments, and to stay colder than the air in warm environments. The temperature at which this

occurs is called "equivalence temperature", which Linacre (1967) measured to be around 34 °C. Linacre hypothesized that the





equivalence temperature can be interpreted as the optimum temperature for plant productivity and growth. It varies between different plant species from 25 °C to 40 °C and averages around 30 °C.

Helliker and Richter (2008) showed that the oxygen isotope ratio ($\delta^{18}$O) of cellulose, which previous studies used as a proxy for the ambient air temperature, exhibits a remarkably constant temperature of 21±2.2 °C ranging from boreal to subtropical regions. This indicates that, even on decadal time scales, the average leaf temperature deviates from the ambient air temperature and tends to stay at a rather constant optimal value, which results in maximal plant productivity. This hypothesis was tested by Michaletz et al. (2016), who collected and analyzed not only the long term $\delta^{18}$O measurements, but also a large number of short-term point measurements of 1504 leaves from 185 different taxa (including the data of Linacre, 1964). Analyses included data from plants that were not water-stressed and predominantly recorded as sunlit. The regression line between the leaf temperature excess and the ambient temperature resulted in an equivalence temperature of 30.1 °C with a fitted slope of the regression line (SRL) of −0.27. Moreover, they concluded that LT "originates from the optimization of leaf traits to maximize leaf carbon gain" regarding plant metabolism; i.e., plants maximize their leaf net carbon gain by increasing carbon assimilation rate while simultaneously decreasing leaf mass per area (Michaletz et al., 2016).

Many aspects of LT have been investigated in recent years, facilitating a better understanding of the thermal and photosynthetic response of the leaf to atmospheric conditions for various leaf functional traits. Rey-Sánchez et al. (2016), e.g., quantified the leaf temperature fluctuations in a tropical forest and developed different empirical approaches to estimate leaf temperature variability by means of air temperature and photosynthetic activity for dry and wet seasons, while Fauset et al. (2018) found different strategies of LT and water use in three tree species in the Atlantic montane forest in Brazil. However, previous studies focus at the leaf scale and short-term observations mostly under water-unstressed conditions. The respective measurements were predominantly taken from sunlit leaves, that merely reflect the conditions at the surface of the canopy. Especially in the case of tall vegetation, the canopy has a complex structure where the shaded leaves, lower or deeper in the canopy, receive less radiation and may behave quite differently than the leaves at the canopy's surface. Thus, while the effect of LT on individual sunlit leaves is well documented, its effect on the canopy as a whole has not yet been studied. To help closing this important knowledge gap, the present study aims to investigate LT in the context of a complex Earth system model (ESM).

Here we use the Max Planck Institute for Meteorology's Earth system model (MPI-ESM; Mauritsen et al., 2019), including a new two-source surface energy balance scheme called "Dual-source Canopy Energy Balance" (CEBa; see section 2). This scheme follows the basic idea of Shuttleworth and Wallace (1985) and Sellers et al. (1986), representing evapotranspiration at specific temperatures from the vegetation and the bare soil, separately. Most importantly, CEBa distinguishes between the temperatures of the leaf, the ambient air, and the ground, allowing to simulate LT with JSBACH. With the help of this adapted model version, we aim to answer the following questions: How do atmospheric conditions and leaf properties influence the leaf's ability to regulate its temperature? And, does the negative correlation between the leaf temperature excess and air temperature – observed at the leaf scale – hold at the canopy, regional and global scale? Finally, different terrestrial ecosystem processes depend on different near surface temperature components; e.g. transpiration and photosynthesis depend on the temperature of the leaf, while bare soil evaporation and decomposition depend on the temperature of the ground. Here we ask





the question whether explicitly representing these individual temperature components affects the simulated terrestrial carbon cycle.

Section 2 provides a detailed description of the CEBa scheme and outlines the experimental setup including the data used in this study. In section 3, the process of LT is analyzed: First, we establish – in a zero-dimensional experiment using the stationary solution of the energy balance – the influence of atmospheric conditions and leaf properties on the leaf's ability to regulate

its temperature. Furthermore, we perform offline experiments, in which the land surface model is driven by observations, for a tropical and temperate forest FLUXNET site (Pastorello et al., 2020) to identify the effect of LT at the canopy scale. Additionally, we perform a global coupled land–atmosphere AMIP (Atmospheric Model Intercomparison Project Gates, 1992) model experiment with CEBa to establish whether the LT signal can be found over a sizeable spatial area and on a decadal time scale in model simulations. Finally, we compare the results to a simulation without the effect of LT on the stomatal control

(i.e., calculated with the air temperature instead of the leaf temperature), to determine whether the consideration of LT affects the simulated terrestrial carbon cycle. In section 4, we summarize our results and discuss the limitations of our study, as well as the need and the potential for future research.

## 2   Model, data, and experiments

This section introduces the CEBa model and describes the design of the zero-dimensional experiment, the single-site FLUX-

NET experiment and the global AMIP experiment.

### 2.1   CEBa

The standard version of JSBACH (hereafter called Classic) and the recently implemented, extended SkIn$^+$ scheme (Heidkamp et al., 2018) both employ a closure of the surface energy balance, where the vegetation is represented by a single layer, based on the big-leaf approach. In these approaches, only one temperature – the surface temperature, corresponding roughly to the

temperature at the displacement height – is used to characterize the canopy air space, the ground under or next to the canopy, and the leaves within the canopy. To be able to represent leaf thermoregulation (LT) with JSBACH, we implemented the CEBa scheme which allows the calculation of four prognostic variables: the leaf temperature $T_{\mathrm{leaf}}$, the ground temperature $T_{\mathrm{grd}}$, the temperature in the canopy air space (CAS) $T_{\mathrm{cas}}$ and the specific humidity in the CAS $q_{\mathrm{cas}}$. The canopy temperature $T_{\mathrm{c}}$ is often described by the temperature of the entire above-ground vegetation, including stems and branches. Since CEBa does

not distinguish between different parts of the biomass in the canopy, in particular the thermodynamic properties of branches, trunks, and leaves, which are all described by the same temperature, we use $T_{\mathrm{c}}$ as a proxy for the leaf temperature $T_{\mathrm{leaf}}$.

Many land surface schemes with a dual-source canopy layer (e.g., Samuelsson et al., 2006) solve the system of energy balance equations diagnostically for $T_{\mathrm{cas}}$ and $q_{\mathrm{cas}}$ without accounting for the heat capacity of the canopy air space, neglecting thermal energy storage as well as changes in enthalpy due to changes in the composition of the CAS. In contrast, we follow

Vidale and Stöckli (2005) and consider heat and water storage in the CAS, which can not be neglected in the case of tall vegetation (Heidkamp et al., 2018). In CEBa, the biomass heat storage depends on the vegetation temperature, while $T_{\mathrm{cas}}$ is





used to calculate the heat storage of the moist air in the canopy. Here the latent heat storage (i.e., heat contained in the humidity in the canopy layer) is no longer calculated by means of the effective surface specific humidity (Heidkamp et al., 2018). It is now calculated more realistically using the specific humidity $q_{\text{cas}}$, representing the moisture content in the CAS.

In Classic and SkIn$^+$, the distribution of the vegetation follows the so-called "land mosaic" configuration, in which the plants are assumed to be clumped in one subgrid area of a grid box (vegetation fraction) and the bare soil occupies the rest (non-vegetation fraction). Both fractions receive the same amount of radiation and the total evaporation is calculated using area-weighted means. In contrast, in CEBa a sparse canopy is used in which the vegetation is distributed uniformly, similar to a savanna-type landscape (Lee, 2018). Here the total evaporation is the sum of transpiration and bare soil evaporation, while

the weighting of their distribution is achieved indirectly by partitioning the net radiation $R_{\text{net}}$ using the so-called "sky-view factor" $\chi$ (Verseghy et al., 1993)

$$\chi = 1 - \left[ f_{\text{veg}} \left( 1 - e^{-0.5\text{LAI}_{\text{eff}}} \right) \right] \tag{1}$$

Here $f_{\text{veg}}$ is the vegetated fraction of the grid box and $\text{LAI}_{\text{eff}}$ is the effective leaf area index taking into account the clumping of vegetation by canopy gaps (Loew et al., 2014). This leads to a radiation flux of the canopy $R_{\text{net,c}}$ and of the underlying ground

$R_{\text{net,g}}$ (Fig. 2). Note that in the following equations the index $c$ pertains to the canopy (which serves in this study analogously for the leaf) and the index $g$ to the ground beneath:

$$R_{\text{net,c}} = \left( 1 - \chi \right) \left[ \left( 1 - \alpha_{\text{c}} \right) S_{\text{in}} + \varepsilon_{\text{c}} L_{\text{in}} \right] + \varepsilon_{\text{c}} (1 - \chi) \left( \sigma T_{\text{g}}^4 - 2\sigma T_{\text{c}}^4 \right)$$
$$R_{\text{net,g}} = \chi \left[ \left( 1 - \alpha_{\text{g}} \right) S_{\text{in}} + \varepsilon_{\text{g}} L_{\text{in}} \right] + \varepsilon_{\text{g}} \left( 1 - \chi \right) \sigma T_{\text{c}}^4 - \varepsilon_{\text{g}} \sigma T_{\text{g}}^4 \tag{2}$$

Here $\alpha$ is the albedo, $\varepsilon$ the surface emissivity, $\sigma$ the Stefan-Boltzmann constant, $L_{\text{in}}$ the incoming longwave radiation and $S_{\text{in}}$ the incoming shortwave radiation. The factor 2 in $R_{\text{net,c}}$ follows the assumption that the canopy layer emits radiation in

two different directions: to the sky and to the soil surface. The radiation budget leads to two different cases of limit value calculation: For a negligible leaf area index (LAI = 0) the sky view factor is maximal ($\chi = 1$), which means that there would be no canopy at all leading to the known solution for bare soil:

$$R_{\text{net,c}} = 0$$
$$R_{\text{net,g}} = \left( 1 - \alpha_{\text{g}} \right) S_{\text{in}} + \varepsilon_{\text{g}} L_{\text{in}} - \varepsilon_{\text{g}} \sigma T_{\text{g}}^4 \tag{3}$$

In contrast, a maximum leaf area index (LAI $\to \infty$) results in a minimum sky view factor of $\chi = 0$ and an opaque canopy layer

that absorbs all incoming shortwave radiation:

$$R_{\text{net,c}} = \left( 1 - \alpha_{\text{c}} \right) S_{\text{in}} + \varepsilon_{\text{c}} \left( L_{\text{in}} + \sigma T_{\text{g}}^4 - 2\sigma T_{\text{c}}^4 \right)$$
$$R_{\text{net,g}} = \varepsilon_{\text{g}} \sigma T_{\text{c}}^4 - \varepsilon_{\text{g}} \sigma T_{\text{g}}^4 \tag{4}$$

Here the soil surface still absorbs and emits longwave radiation in interaction with the overlying dense canopy layer.

 Figure 2 illustrates the energy and water exchange between the canopy, the ground, the canopy air space, and the lowest atmospheric level (LAL) of the new CEBa system, including the temperatures, humidities, and resistances, respectively. The

sensible and latent heat fluxes are parameterized using the common resistance formulation.





**Figure 2. Principal sketch of the dual-source canopy energy balance scheme CEBa:** Depicted are the radiative heat exchanges (yellow for shortwave radiation, orange for longwave radiation), temperatures (red), specific humidities (blue), energy fluxes (red arrows), and water fluxes (blue arrows), including the resistances (black), where $\chi$ is the sky-view factor, $L_{\text{in}}$ the incoming longwave radiation, $S_{\text{net}}$ the net shortwave radiation, $\varepsilon$ the surface emissivity, $\sigma$ the Stefan-Boltzmann constant, $T_{\text{air}}$ the air temperature, $T_{\text{leaf}}$ the leaf temperature, $T_{\text{grd}}$ the ground surface temperature, $T_{\text{cas}}$ and $q_{\text{cas}}$ the temperature and specific humidity in the canopy air space, $q_{\text{air}}$ the specific humidity of the air, $q_{\text{sat}}$ the saturated specific humidity at the respective temperature, $r_{\text{atm}}$ the atmospheric resistance, $r_{\text{b}}$ the bulk resistance, $r_{\text{d}}$ the aerodynamic resistance between ground and CAS, $r_{\text{c}}$ the canopy resistance, and $\text{RH}_{\text{up}}$ a nonlinear function depending on the relative humidity of the uppermost soil layer.

There are three different pairs of sensible and latent heat fluxes that are connected to each other through the CAS. The first is the exchange of energy and water between the CAS and the atmosphere:

$$H_{\text{atm}} = \rho c_p \frac{T_{\text{air}} - T_{\text{cas}}}{r_{\text{atm}}} \qquad LE_{\text{atm}} = \rho L_{\text{v}} \frac{q_{\text{air}} - q_{\text{cas}}}{r_{\text{atm}}} \tag{5}$$




where $\rho$ is the density of humid air, $c_p$ the specific heat capacity of air, $L_v$ the specific enthalpy of vaporization, $T_{air}$ and $q_{air}$
the temperature and specific humidity of the air at the LAL, $T_{cas}$ and $q_{cas}$ the temperature and specific humidity of the air in
the CAS, and $r_{atm}$ the atmospheric aerodynamic resistance, which is the inverse product of wind speed and drag coefficient.
The drag coefficient represents a measure of the turbulence strength depending on the roughness of the underlying surface and
the atmospheric stratification. The second pair is the sensible and latent heat flux between the CAS and the canopy biomass
(for simplicity's sake snow and interception evaporation is not considered here):

$$H_c = \rho c_p \frac{T_{cas} - T_c}{r_b} \qquad LE_c = \rho L_v \frac{q_{cas} - q_{sat}(T_c)}{r_b + r_c} \tag{6}$$

where $T_c$ is the temperature of the canopy surface (here a proxy for the leaf temperature $T_{leaf}$), $q_{sat}$ the saturated specific
humidity at the respective temperature, $r_b$ the aerodynamic resistance between the canopy and the CAS, and $r_c$ the canopy re-
sistance. The latter describes the additional resistance that water vapor molecules encounter when moving through the stomatal
openings of leaves. It is the reciprocal of the stomatal conductance, which depends on the assimilation rate and hence on the
carbon flux through the stomata (Farquhar and Sharkey, 1982). It is scaled by the LAI (leaf area index) to achieve the transition
from leaf scale to canopy scale. In addition, it is modified by a water stress factor that depends on the saturation of the root
zone. Since photosynthesis depends on the leaf temperature, in CEBa, $r_c$ depends on the leaf temperature instead of the air
temperature of the LAL. The resistance between the leaf and the canopy air space $r_b$ (also called bulk resistance) is a function
of the LAI, the leaf width, and the wind speed at the top of the canopy $u_{cas}$ and describes the additional resistance induced
by the turbulence in the leaf boundary layer (Choudhury and Monteith, 1988). It is modified with a free convection correction
according to Sellers et al. (1986) depending on $T_c$, $T_{cas}$ and the LAI. The last pair is the sensible and latent heat flux between
CAS and the ground surface under the canopy:

$$H_g = \rho c_p \frac{T_{cas} - T_g}{r_d} \qquad LE_g = \rho L_v \frac{q_{cas} - \mathrm{RH}_{up} q_{sat}(T_g)}{r_d} \tag{7}$$

where $T_g$ is the temperature of the ground surface, $r_d$ the aerodynamic resistance between the ground surface and the CAS,
and $\mathrm{RH}_{up}$ a nonlinear function depending on the relative humidity of the uppermost soil layer. The latter serves as a resistance
for bare soil evaporation. The parameterization of $r_d$ follows a more complex approach depending on the vegetation height
$z_{veg}$, the zero plane displacement height $d$ (again a function of $z_{veg}$), and the wind speed at the top of the vegetation $u_{cas}$. It is
based on the general equation of an aerodynamic resistance for momentum transfer between the soil surface and the sink for
momentum in the vegetation under neutral conditions (Choudhury and Monteith, 1988). We also apply a stability correction
for unstable cases following Sellers et al. (1986). For the estimation of $u_{cas}$, we use an approach according to Xue et al. (1991).
The basic idea is the concept of a roughness layer defined by a so-called transition height $z_{trans}$ (see Appendix B).





Considering these expressions, we obtain a system of four coupled equations:

$$C_c \frac{\partial T_c}{\partial t} = H_c + LE_c + R_{\text{net,c}} + \beta F_{CO_2}$$

$$0 = H_g + LE_g + R_{\text{net,g}} + G$$

$$C_T \frac{\partial T_{\text{cas}}}{\partial t} = H_{\text{atm}} - H_c - H_g$$

$$L_q \frac{\partial q_{\text{cas}}}{\partial t} = LE_{\text{atm}} - LE_c - LE_g$$

(8)

Here $C_c$ and $C_T$ are the area-specific heat capacities of the canopy and the CAS, respectively, $L_q$ is the area-specific enthalpy

of vaporization for water, $G$ is the ground heat flux, $F_{CO_2}$ is the net $CO_2$ flux and $\beta = 10.884 \cdot 10^6$ J/kg is a conversion factor corresponding to the energy required to incorporate $1 \, \text{mol} \, CO_2$ by carbohydrate bonds through the process of photosynthesis. We assume an infinitesimally thin surface at the ground, which allows to include the ground heat storage by the ground heat flux $G$. The nonlinear terms within the system of equations in CEBa are linearized using a first-order Taylor approximation and the time derivatives are discretized. The resulting linear system of four prognostic variables is solved by matrix inversion

and iteration.

## 2.2 Experiments and data

For the first experiment, we do not run the full ESM or even the complete land-surface component. Rather, we solve the system of energy balance equations of the CEBa model for different air temperatures ranging from $-20 \, °C$ to $60 \, °C$. Here we assume an unlimited soil water availability for transpiration (no water stress), as well as for bare soil evaporation ($\text{RH}_{\text{up}} = 1$). The

surface parameters were chosen to represent a forest with tall vegetation – i.e., $z_{\text{veg}} = 30 \, \text{m}$, $z_0 = 4.8 \, \text{m}$, $\text{LAI} = 4.5 \, \text{m}^2/\text{m}^2$, where $z_{\text{veg}}$ is the vegetation height, $z_0$ the roughness length, and LAI the leaf area index. Furthermore, we prescribe the specific humidity of the air, as well as the wind speed and the incoming solar radiation. By varying the latter, we can analyze how the equivalence temperature ($T_{\text{eq}}$) and the slope of the regression line (SRL) depend on the relative humidity, incoming solar radiation, canopy resistance, and aerodynamic leaf resistance. Although the leaf temperature excess as a function of air

temperature represents a nonlinear curve, we approximate it by a linear function, in order to describe the effect of LT simply by the SRL and $T_{\text{eq}}$.

In this experiment, we calculate the stationary solution as a function of the air temperature to determine $T_{\text{eq}}$ and the SRL describing the negative correlation due to LT. This steady-state experiment can be interpreted as a zero-dimensional experiment since the solution neither depends on space nor on time and there are neither heat nor water mass storages. In this case, the heat

fluxes between the CAS and the LAL are the sums of the ground and canopy heat fluxes. Thus, the third and fourth equation of (8) simplify to

$$H_{\text{atm}} = H_c + H_g$$

$$LE_{\text{atm}} = LE_c + LE_g$$

(9)



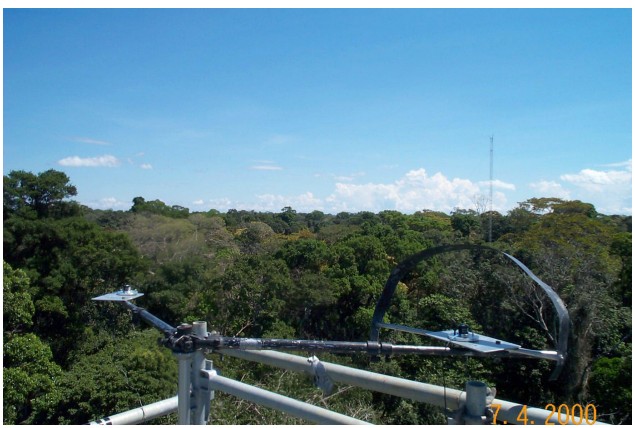

**Figure 3. Location of the FLUXNET tower site** located in the tropical evergreen broadleaf forest, Brazil, photo taken from `https://fluxnet.fluxdata.org` (last access 01 Oct 2020, URL: http://sites.fluxdata.org/files/media/images/br-sa1-primary_forest_tower.jpg)

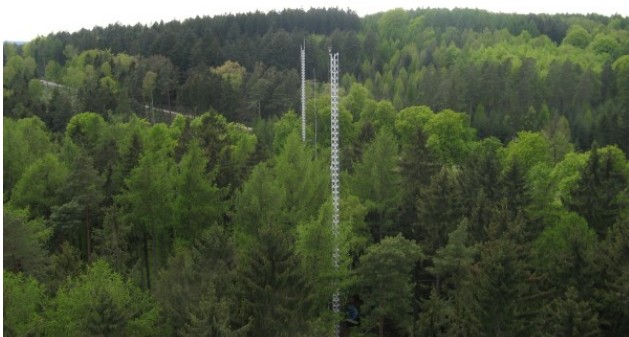

**Figure 4. Location of the FLUXNET tower site** located in the evergreen needleleaf forest in Tharandt, Germany, photo taken from `https://tu-dresden.de` (last access 01 Oct 2020, URL: https://tu-dresden.de/bu/umwelt/hydro/ihm/meteorologie/ressourcen/bilder/forschung/projekte/turbefa/bilder/aufbau_13?lang=en)

To obtain the total net radiation the partial radiative fluxes have to be weighted by the sky view factor $\chi$:

$$R_{\mathrm{net}} = (1 - \chi)R_{\mathrm{c}} + \chi R_{\mathrm{g}} \tag{10}$$

Note that it is debatable whether air temperature measurements resemble $T_{\mathrm{cas}}$, the temperature next to (or even under) the vegetation within the CAS, or $T_{\mathrm{air}}$, the air temperature above the canopy. In the respective analysis, the leaf temperature excess is defined as the temperature difference between $T_{\mathrm{leaf}}$ and $T_{\mathrm{air}}$.





To address the second scientific question, i.e. whether a negative correlation of leaf temperature excess and air tempera-
ture emerges under realistic conditions at the canopy scale, we performed two offline single-site experiments with JSBACH
including CEBa, using 30-minute measurement data from a FLUXNET tower in Brazil's tropical forest (Goulden, 2016) and
the temperate Tharandt forest in Germany (Bernhofer et al., 2016). Both datasets are part of the "FLUXNET2015 Dataset"
(Pastorello et al., 2020). In an offline experiment, the LSM is decoupled from its host model and forced by observation data (pre-
cipitation, air pressure, air temperature, specific humidity, and wind, as well as shortwave and longwave incoming radiation).
Both are forest sites with a dense canopy and a large vegetation height. The tropical site (ID: BR-Sa3, $-3.0°$ N, $-55.0°$ E,
100 m elevation) is located in an evergreen broadleaf forest (Fig. 3) with a vegetation height of around 35-40 m, while the
measurement height is at 64 m. The average annual temperature is 26.1 °C and the annual mean precipitation is 2044 mm.
The Tharandt FLUXNET tower (ID: DE-Tha, 51.0° N, 13.6° E, 385 m elevation) is located in an evergreen needleleaf forest
(Fig. 4) with a vegetation height of around 30 m, while the measurement height is at 42 m (Delpierre et al., 2009). The annual
mean temperature is 8.2 °C and the annual mean precipitation is 843 mm. For both locations, the surface and soil settings
are extracted from the standard settings of the respective JSBACH grid box and adapted to the properties of the site. For the
tropical site, data are only available from September 2001 to October 2003. Therefore, these years are randomly resampled
for a 10-year initialization period to ensure equilibrium with respect to the temperature and moisture content in the soil. Since
the Tharandt site provides a long period of data, the data from 1996 to 2005 are used for the spin-up of the model and the
data from 2006 to 2014 for the actual experiment. Since photosynthesis occurs only under the presence of sunlight, we analyze
only daytime values (defined by inclination angles greater than 10°) of simulated leaf and air temperatures in the following
experiments.

To investigate the effect of LT on the global scale, we performed a set of coupled simulations with the MPI-ESM at T63
resolution (i.e. 1.9°), covering the years 1979 to 2008. The simulations follow the AMIP protocol (Gates, 1992), with the sea
surface temperatures and the sea-ice extent being prescribed. Technically, it is possible to run the CEBa scheme for shallow
or even no vegetation, such as deserts or glaciers (the latter with an average vegetation height of $z_{\mathrm{veg}} = 0$). However, the
simulations with CEBa for these extremes produce implausible results. Therefore, the CEBa scheme is used only in grid boxes
with an average vegetation height greater than 5 m. The resulting mask is depicted in Fig. 5. For all other land points, the
SkIn$^+$ scheme is used and the grid boxes are excluded from the following analysis. Both the offline and coupled simulations
are run with a time step of 7.5 min.

## 3 Results

In this section the process of leaf thermoregulation (LT) is analyzed as follows: First, we asses the influence of atmospheric
conditions and leaf properties on the leaf's ability to regulate its temperature. Secondly, we perform offline experiments with
JSBACH including the CEBa scheme for a tropical and a temperate forest site to identify the effect of LT at the canopy scale
under realistic conditions. Finally, we perform a global coupled model experiment with CEBa to establish whether a negative



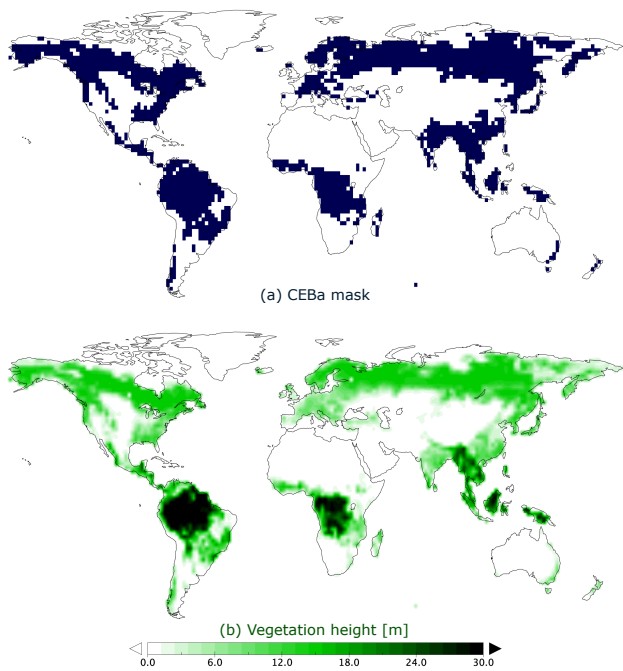

**Figure 5. CEBa 5-m vegetation height mask** indicating for which grid cells the CEBa scheme is used (a) and average vegetation height in JSBACH (b)

correlation between leaf temperature excess and ambient air temperature can be found over a sizeable spatial area and on a decadal time scale in model simulations.

### 3.1 Steady state experiment

In the temperature range between $0\,^\circ$C and $55\,^\circ$C, the system of energy balance equations of the new CEBa scheme (Eq. 8) reproduces the negative correlation between leaf temperature excess $\Delta T$ and ambient air temperature that is characteristic
of LT (Fig. 6, see also Appendix C). For low air temperatures, where transpiration is negligible, the leaf is warmer than the surrounding air because the incoming shortwave and longwave radiation is absorbed and transformed into internal energy, resulting in an increase of the leaf surface temperature. In equilibrium, this energy is released in the form of fluxes of outgoing longwave radiation and sensible heat, which transfer energy from the leaf surface to the air and compensate for the radiative forcing. For higher temperatures, $\Delta T$ decreases almost linearly with increasing air temperatures, which is a result of the non-
linear temperature-dependency of the latent heat flux. The sensible and latent heat fluxes are mainly driven by the temperature and moisture gradient between leaf and air and for a linear temperature increase, the saturated specific humidity increases exponentially following Clausius–Clapeyron theory. At a (comparatively) constant net radiation and relative humidity in the canopy and assuming water-unstressed plants, this results in an energetic equilibrium, in which the latent heat flux increases with rising temperatures, whereas the sensible heat flux decreases. Above a certain temperature, the latent heat flux exceeds





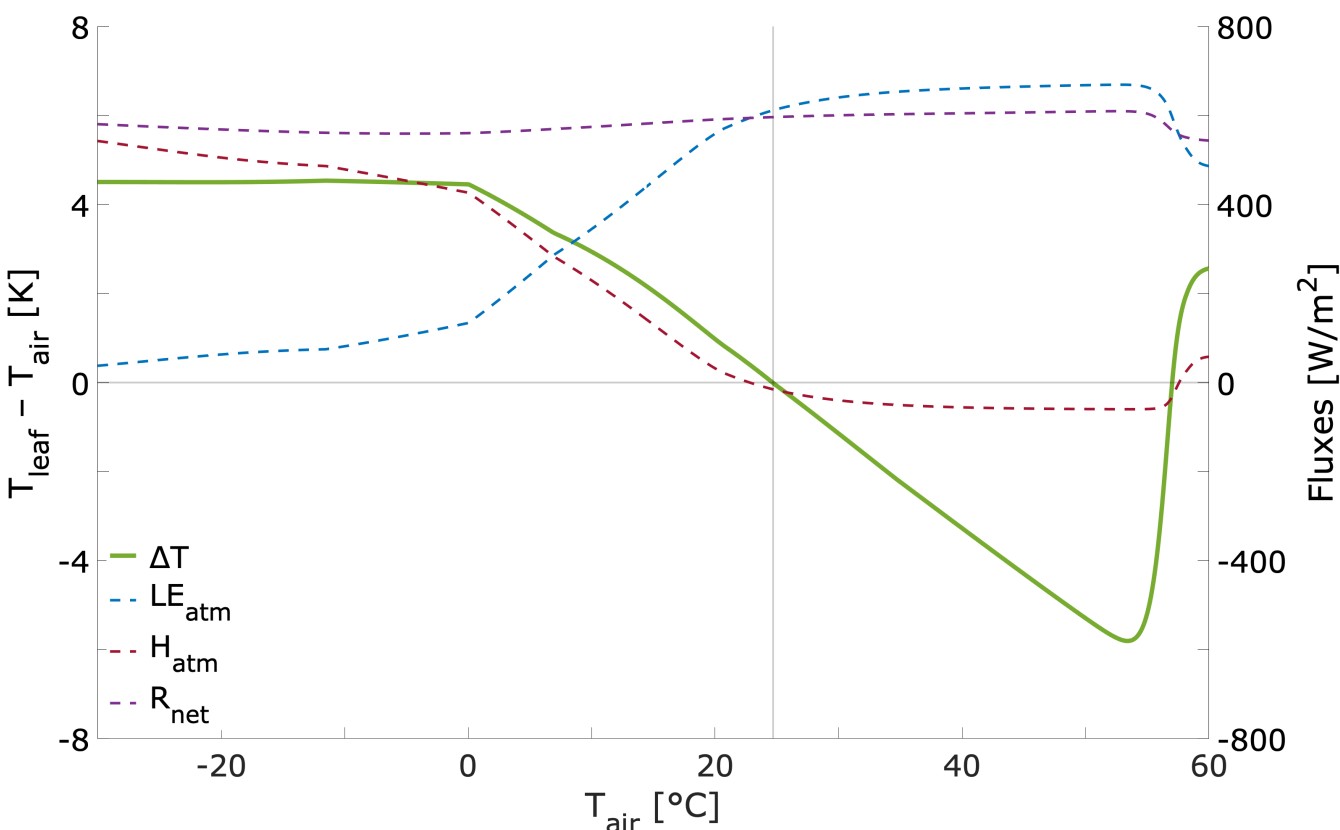

**Figure 6. Steady-state solution of the CEBa energy balance:** Plotted are the leaf temperature excess (solid green line, left y-axis), the total latent heat flux (dashed blue lines, right y-axis), the total sensible heat flux (dashed red lines, right y-axis) and the net radiation (dashed violet lines, right y-axis) as a function of the air temperature at the reference level. The solution is valid for: $S_{in} = 800 \, \mathrm{W/m^2}$, $\mathrm{RH} = 50\%$ and $v = 3 \, \mathrm{m/s}$

the net radiation resulting in the oasis effect, due to which the sensible heat flux becomes negative and the leaf absorbs energy from the surrounding air. In the example depicted in Fig. 6, this equivalence temperature ($T_{eq}$) is reached at 24.7 °C. At temperatures of about 55 °C, $\Delta T$ increases rapidly due to a high-temperature inhibition (implemented in the calculation of the stomatal control) that limits transpiration under temperature extremes.

The slope of the regression line (SRL) and $T_{eq}$ – which can be used to characterize the effect of LT – depend on atmospheric conditions, such as incoming radiation, but also on the characteristics of the canopy, such as the canopy resistance (Fig. 7). $T_{eq}$ increases for higher radiative fluxes and for higher humidity, with the $T_{eq}$ dependency on the incoming radiation following a logarithmic function and an exponential function for relative humidity. This implies that variations in solar radiation are especially important when the solar incoming radiation is low (lower than $400 \, \mathrm{W/m^2}$), while variations in the air's moisture content are important when the relative humidity is high (higher than 80 %). The SRL increases with saturation but decreases



**Figure 7. Variations of the leaf thermoregulation characteristics** in terms of equivalence temperature ($T_{eq}$, solid lines, each left y-axis) and the slope of the regression line at $T_{eq}$ (dashed-lines, each right y-axis) as a function of incoming solar radiation $S_{in}$ (a), relative humidity RH (b, for both using the simulated resistances), canopy resistance $r_c$ (c), and aerodynamic resistance $r_b$ (d).

with rising radiative fluxes, indicating LT to be more effective at higher radiative fluxes and in dryer atmospheric conditions. Here the SRL displays an almost linear dependency on incoming solar radiation and relative humidity, indicating a similar sensitivity at high and at low values. In contrast, changes in canopy resistance $r_c$ and aerodynamic leaf resistance $r_b$ exhibit a notable influence on the LT curve almost exclusively at low values of $r_b$ and $r_c$. Here $T_{eq}$ increases logarithmically with increasing resistances for both $r_c$ and $r_b$. The SRL is substantially affected by changes in $r_c$ and $r_b$ when assuming a high incoming solar radiation, while for lower radiative fluxes the SRL is largely independent of the resistances.





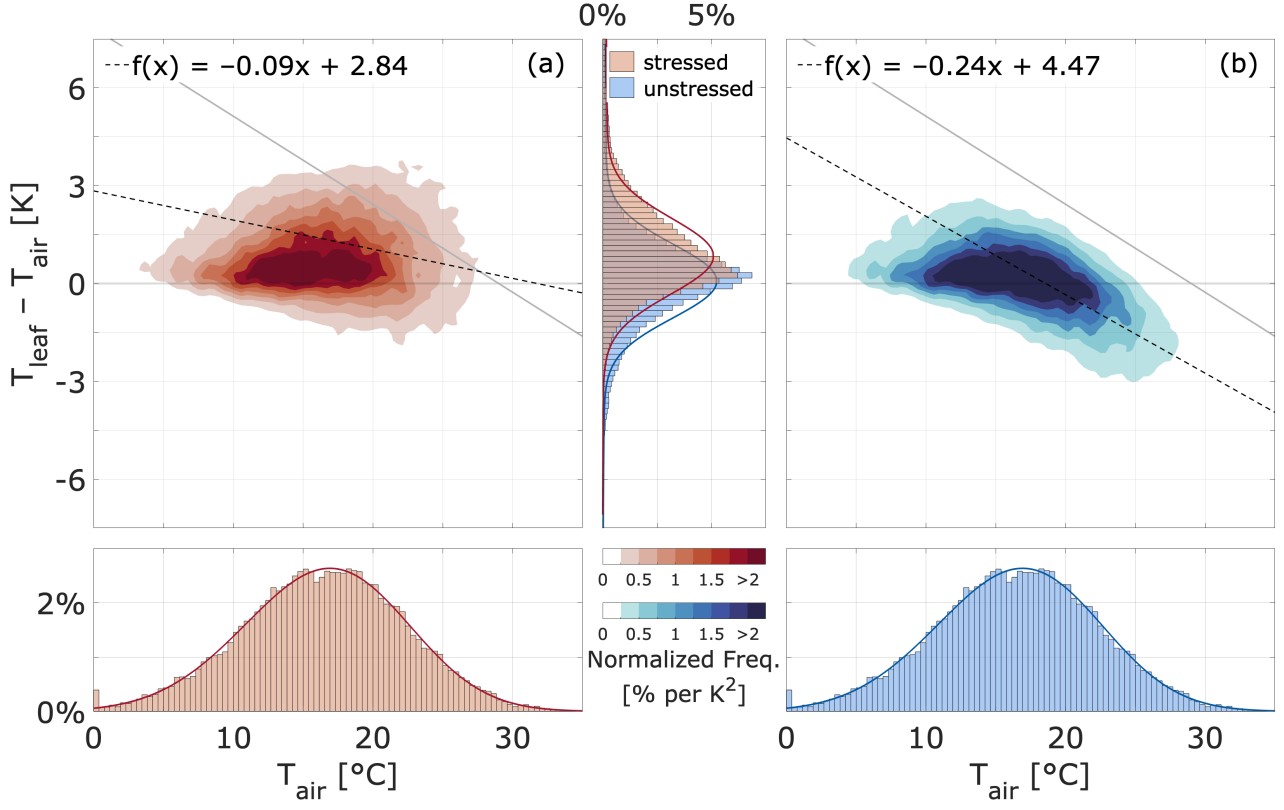

**Figure 8. Temperate Zone: Two-dimensional probability function of daytime leaf temperature excess and air temperature.** Data are from the vegetation period (Apr–Sep, 2006–2014) of the offline experiment for the FLUXNET site located in the temperate needleleaf Tharandt forest in Germany: Reference experiment (a) (stressed, red) and water-unlimited case (b) (unstressed, blue). Plotted are instantaneous values of the model output with 7.5 min time step. The dashed black line is the linear regression through all data of the model output; the gray solid line depicts the regression line derived in the study of Michaletz et al. (2016). The brightness of the colors indicates how many percents of the data fall within an "area" of $1\,\text{K} \times 1\,\text{K} = 1\,\text{K}^2$.

## 3.2 FLUXNET Sites

The above results show that the strength of the LT effect – given by the SRL – is in large parts determined by the atmospheric conditions, with a strong LT being possible only in case of a dry atmosphere and a strong solar flux. This raises the question whether the negative correlation between leaf temperature excess and air temperature also constitutes a robust feature under realistic atmospheric conditions. To adress this question, we use JSBACH with the model being forced using observations from two FLUXNET sites located in the temperate needleleaf Tharandt forest in Germany and in a tropical evergreen forest in Brazil.





At the Tharandt site, the daytime air temperatures during the vegetation period range between 0 °C and 35 °C and follow a Gaussian distribution (Fig. 8a). The leaf temperature excess shows a sharp drop in frequency for negative temperature differ-
ences and the two-dimensional probability function indicates a larger spread towards positive (3.7 K for frequencies larger than $0.25\,\%/\mathrm{K}^2$) than towards negative leaf temperature excesses ($-1.6\,\mathrm{K}$ for frequencies larger than $0.25\,\%/\mathrm{K}^2$). Nevertheless, the regression line between leaf temperature excess and air temperature exhibits a negative correlation, showing that the effect of LT is present also at the canopy scale under observed atmospheric conditions. Here $T_{\mathrm{eq}}$ is equal to 31.7 °C and similar to the 30.1 °C reported by Michaletz et al. (2016). However, the SRL ($-0.1$) is significantly smaller than the SRL of $-0.27$ found
by Michaletz et al. (2016) and the simulated LT is constantly weaker than indicated by observations. One potential reason for the mismatch between the simulated and the observed SRL is the water-stress level of the vegetation. While the simulations account for limitations due to water-availability, Michaletz et al. (2016) investigated water-unstressed plants. When neglecting water stress in the simulations, the distribution of $\Delta T$ shifts towards negative temperature differences – the largest significant ($> 0.25\,\%/\mathrm{K}^2$) negative differences now amount to $-3\,\mathrm{K}$ – and the simulated SRL ($-0.24$) agrees well with the observations
(Fig. 8b). However, the stronger evaporative cooling reduces the simulated $T_{\mathrm{eq}}$ to 18.6 °C, which is around 12 K below the value reported by Michaletz et al. (2016).

In general, the simulated leaf temperatures are substantially lower than those reported by Michaletz et al. (2016) resulting in an almost constant offset between the simulated and observed regression lines in the case of unstressed vegetation (Fig. 8b). This is mainly due to focus and scale differences between the observations and the simulations. Most temperature measure-
ments target individual leaves at the surface of the canopy, while the CEBa scheme covers the entire extent of the canopy layer. Thus, while the observations predominantly pertain to the temperatures of the warmer leaves, which receive direct sunlight, the CEBa scheme estimates the average over the entire canopy biomass, including the shaded parts of the canopy, which are substantially cooler.

At the tropical site, the simulated effect of LT is not only weaker than the observed values, but the leaf temperature excess
even shows a positive correlation (SRL = 0.15) with the air temperatures (Fig. 9a). Similar to the Tharandt site, one reason for the high SRL is the increasing water stress at higher temperatures, and neglecting the limitations due to soil water availability results in a negative correlation between $\Delta T$ and $T_{\mathrm{air}}$ (Fig. 9b). However, even under unstressed conditions, the SRL ($-0.18$) is substantially higher than the observed values ($-0.27$) or the SRL simulated for the Tharandt site ($-0.24$), indicating a weaker LT effect in tropical forests. Here the reason for the weak LT lies in the prevailing high relative humidity in the atmosphere,
which limits the potential for an evaporative cooling of the leaf. While in the unstressed case the average relative humidity at the Tharandt site was 62 %, the tropical site featured a relative humidity of 71 %. Thus, while we find that the strength of LT at the canopy scale under realistic atmospheric conditions is comparable to observations at the leaf scale, this is only the case for dry atmospheric conditions and under water-unstressed conditions. Once limitations due to soil water availability are accounted for, the negative correlation between $\Delta T$ and $T_{\mathrm{air}}$ becomes weaker and even becomes positive for high levels of
relative humidity.





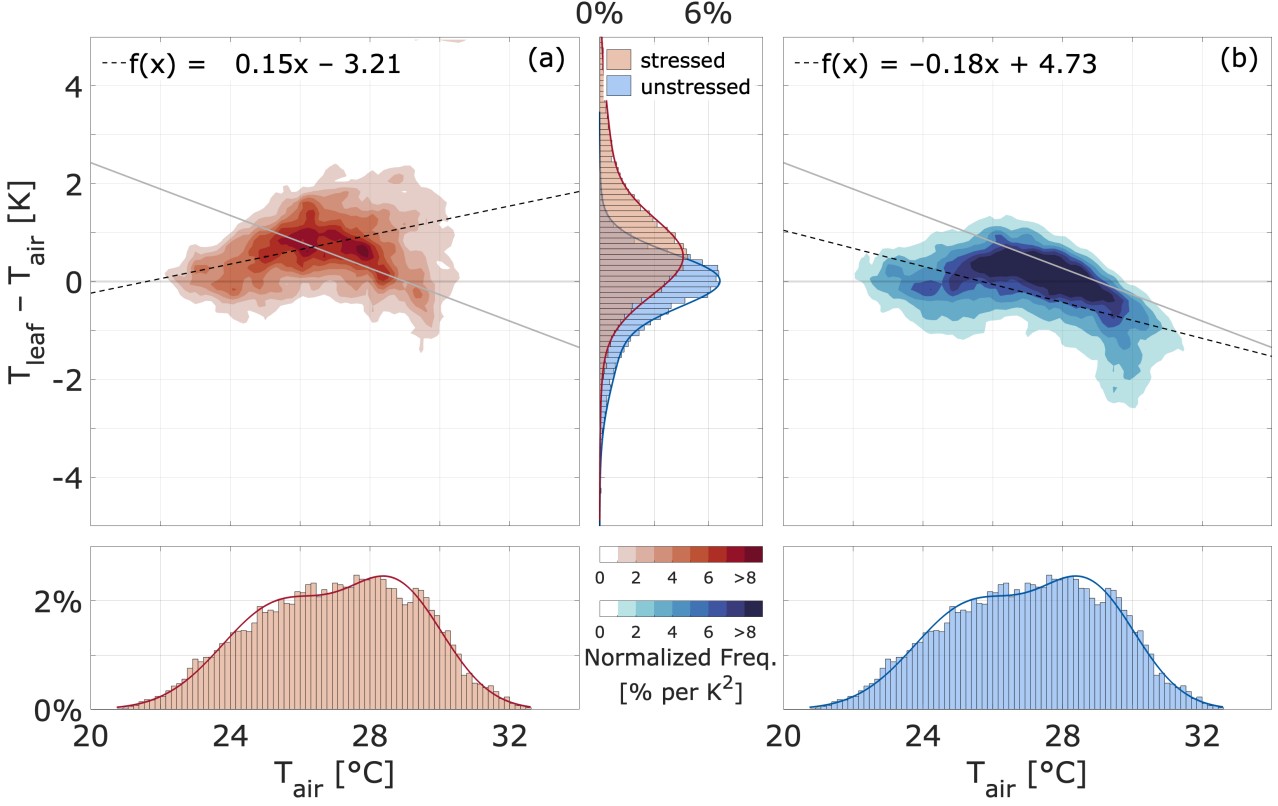

**Figure 9. Tropics: Two-dimensional probability function of daytime leaf temperature excess and air temperature.** Data are from the offline experiment (September 2001 to October 2003) for the FLUXNET site located in a tropical evergreen forest in Brazil: Reference experiment (a) (stressed, red) and water-unlimited case (b) (unstressed, blue). Plotted are instantaneous values of the model output with 7.5 min time step. The dashed black line is the linear regression through all data of the model output; the gray solid line depicts the regression line of the findings of Michaletz et al. (2016)

## 3.3 Global AMIP experiments

The positive correlation between $\Delta T$ and $T_{\mathrm{air}}$ is not unique to the investigated site in the tropics but we also find it to be the dominant long-term signal at the global scale. In an AMIP experiment, the 30-year mean daytime leaf temperature during the vegetation period (Apr–Sep) is consistently larger than the corresponding air temperature, with the smallest $\Delta T$ amounting to

0.5 K. Here we assume that a space-for-time-substitution is justified. This means that the variations of the data points (30-year averages) in Fig. 10, which result due to variations in space at different grid points, can be compared with those data points measured at a single site that are a function of time at a fixed location. Large $\Delta T$ are predominantly simulated in grid cells that also feature high air temperatures, which leads to a positive correlation between leaf temperature excess and air temperature, with a SRL of about 0.04 (Fig. 10, red circles).



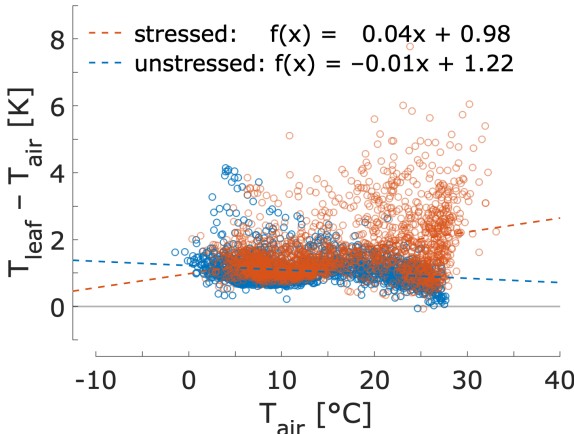

**Figure 10. Scatter plot of the daytime leaf temperature excess and air temperature based on 30-year means.** Data are derived from a long-term global AMIP experiment over 30 years (1979–2008): Reference experiment (stressed, red) and water-unlimited case (unstressed, blue). The dashed lines are the regression lines. Considered are only those grid points in which the CEBa scheme is used to solve the surface energy balance. Daytime values are defined as an inclination angle greater than $10°$.

Similar to the site-level experiments, the positive correlation between $\Delta T$ and $T_{air}$ can be explained by limitations due to water-availability, where a larger water stress is predominantly simulated for grid cells with higher air temperatures. An idealized experiment without water-stress (where unlimited soil water availability is assumed) produces a negative correlation between $\Delta T$ and $T_{air}$, with a SRL of around -0.01 (Fig. 10, blue circles). Here $\Delta T$ constitutes predominantly positive values, which is in good agreement with the observations. However, these positive $\Delta T$ are significantly smaller than the short-term

measurements. Negative temperature difference are hardly present, but a slight negative trend is visible at air temperatures of about $30\,°C$, which would agree with the $T_{eq}$ reported by Michaletz et al. (2016). However, the model does not even simulate 30-year means of daytime air temperature higher than $30\,°C$. This leads to a significantly smaller SRL than the values derived in the site-level experiments of about -0.01. One reason for this feature could be caused by a negative land-atmosphere-feedback, which is present in the AMIP experiment, but not in the site level simulations, where the air temperature and specific humidity

at the lowest atmospheric model level are prescribed based on observations. In the AMIP experiment, the evaporative cooling does not only reduce the leaf temperature but also affects the atmospheric temperatures, diminishing the overall effect on $\Delta T$.

    The theory of LT postulates that leaves adjust their temperature to maximize the carbon gain of the plant, but the respective increase in productivity has not yet been assessed at the global scale. In the following we compare the gross primary productivity (GPP) of the above AMIP simulations with that of an additional set of simulations in which the GPP and the canopy resistance

are simulated based on air temperatures. Here we analyze the correlation between the GPP (in $\mu mol\,CO_2/(m^2 s)$) calculated based on air temperatures $GPP(T_{air})$ and the difference $\Delta GPP = GPP(T_{leaf}) - GPP(T_{air})$ for the water-stressed and water-unstressed case, with positive values of $\Delta GPP$ indicating an increase in GPP due to the consideration of LT. Both the stressed and unstressed experiment show a general increase in GPP with regression curves above the x-axis (Fig. 11). However, while





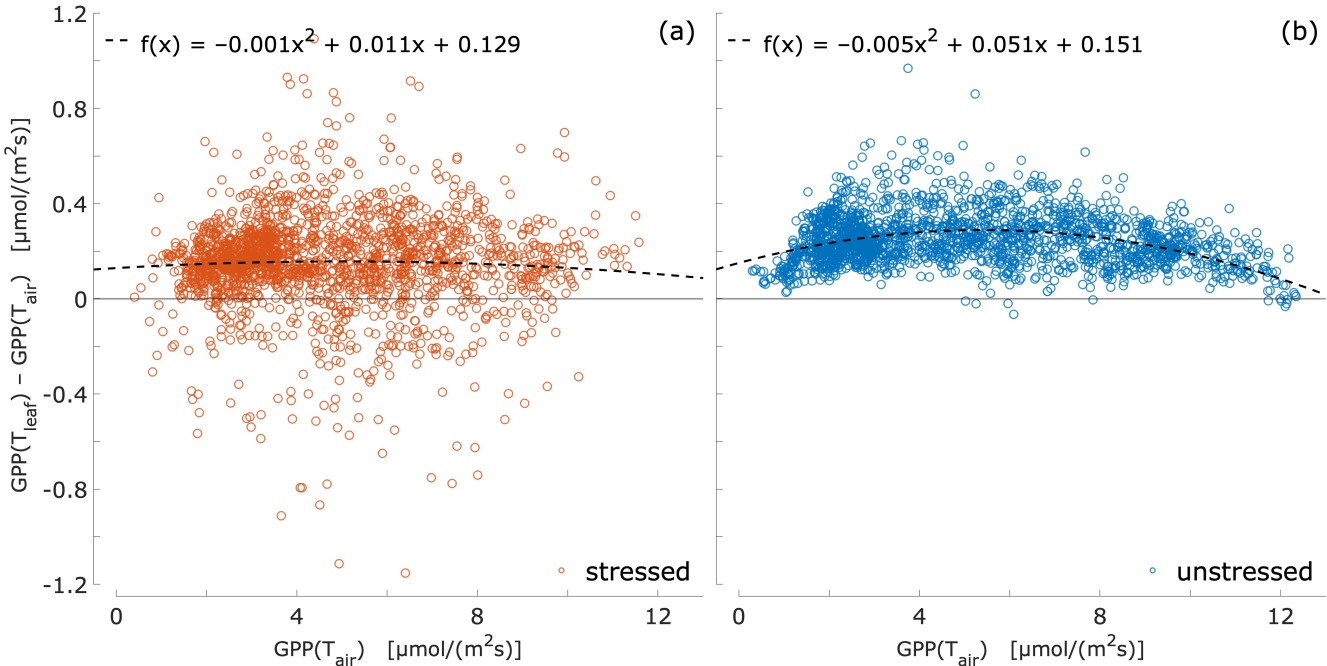

**Figure 11. Scatter plot of 30-year means of gross primary productivity (GPP) calculated using the air temperature GPP($T_{\mathrm{air}}$) and the difference in GPP** calculated using the leaf temperature or the air temperature, respectively, $\Delta$GPP = GPP($T_{\mathrm{leaf}}$) − GPP($T_{\mathrm{air}}$) in $\mu$mol CO$_2$/(m$^2$s) for the reference (water-stressed, red) and idealized (water-unstressed, blue) experiment. Plotted is the simulated model output for CEBa grid cells for the period 1979 to 2008.

the unstressed experiment shows almost exclusively positive values, the stressed experiment simulates a negative $\Delta T$ in several

cases. In general the simulated GPP differences are rather small and the effect of LT appears to be strongest for GPP rates of around 4 to 8 $\mu$mol CO$_2$/(m$^2$s) with a maximum of the regression curve of 0.28 $\mu$mol CO$_2$/(m$^2$s) at 5.1 $\mu$mol CO$_2$/(m$^2$s). Overall, using the leaf temperature instead of the air temperature to calculate the canopy resistance and GPP leads to an increase in the 30-year global GPP mean of around 3 % for the water-stressed experiment and 5 % for the unstressed case. Thus, our results support the theory that the temperature regulation of the leaf constitutes a strategy to maximize the carbon gain of

the plant.

## 4   Discussion and Conclusion

This study constitutes the first investigation of the effect of leaf thermoregulation (LT) at larger spatial and temporal scales. We demonstrated that the characteristic correlation between leaf temperature excess and air temperature is not only a feature of individual leaves but can also be found at the canopy scale. Here our model simulated a negative correlation on the site-

level in case of a temperate forest in Germany. However, the strength of LT – in the form of the slope of the regression line





(SRL) – was underestimated in comparison to the SRL observed at the leaf scale. For a site in the tropics, we even found a positive correlation, which appears to contradict the theory of LT. This positive correlation is not limited to the tropics but constitutes the dominant dependency on the global scale. Here we could show that it is largely the result of simulated moisture limitations, as water stress predominantly increases with rising temperatures and increasingly limits the evaporative cooling of the leaves. In idealized experiments with water-unstressed plants, we simulate the observed SRL both at the temperate- and at the tropical site and the correlations between leaf temperature excess and air temperature is negative also on the global scale. The results on global scale suggest that leaf thermoregulation does not substantially affect the surface or near-surface temperatures on climatological time scales. However, despite the rather weak signal in leaf temperature excess on decadal time scales the global 30-year GPP mean increases up to 5 %.

Our results indicate that the leaf temperature excess is substantially affected by the prevailing soil moisture conditions. The representation of water stress in JSBACH has not yet been evaluated explicitly and could still contain uncertainties, e.g., due to biases in the soil moisture content. Moreover, there are potential water limitations that could control the canopy resistance, possibly, due to missing processes. The temperature dependence for the calculation of the stomatal conductance as well as for the photosynthesis rate is assumed to follow an Arrhenius-type temperature dependence. This means in the context of JSBACH: the higher the temperature, the higher the stomatal conductance or the larger the GPP rate (up to a certain temperature where a high-temperature inhibition sets it to zero). What is missing so far is a parameterization of a process that artificially regulates the stomata to the plant's temperature optimum.

We have shown that the effect of LT (i.e., the amplitude of the leaf temperature excess) depends to a large extent on the relative humidity of the ambient air and the increasing solar insolation. Regarding the latter, it is crucial to estimate the distribution of radiation within the canopy layer correctly, i.e., how much radiation is absorbed or reflected from the leaf and how much is transmitted to the ground surface. In this context, studies have shown that leaves have the ability to reduce or enhance the absorption of solar radiation, e.g., by increasing the leaf or needle density, by rotating the leaf angle, or by reflective leaf hairs (Helliker and Richter, 2008).

We found that, especially during midday, the leaf temperature is almost always higher than the air temperature since it is completely illuminated. In this case, if the leaf temperature were lower than the air temperature at high solar radiation, this would represent a state with stable stratification, which is not realistic in convective conditions during the day. To avoid that, only a large heat flux from the ground surface could compensate that imbalance. Thus, it is questionable whether LT is even possible for extremely dense or closed canopy layers (e.g., in the tropics). In that regard, the implementation of a flux aggregation method to solve the energy balance equation (Best et al., 2004; de Vrese and Hagemann, 2016) could provide a more realistic representation of the canopy layer.

Our results demonstrate that future studies need to consider the vertical distribution of leaf temperatures regarding LT theory to distinguish correctly between leaf and canopy scale. In this context, a multilayer model is probably more appropriate as it can use parameters measured at the leaf scale and at different heights to calculate the vertical temperature gradient in the canopy layer (Dai et al., 2004). Luo et al. (2018) suggest to avoid the big-leaf approach and advocates the use of a two-leaf


scheme, which determines firstly the transpiration of sunlit and shaded leaves separately, and secondly, upscales the results from the leaf to the canopy scale.

    For exploring the role of the leaf temperature in the canopy, we made a first approach to investigate the effect of LT at the canopy scale using an ESM. We show that we are indeed able to detect a negative correlation between leaf temperature excess and air temperature, but its magnitude is underestimated when compared to observations found at the leaf scale. This finding

raises the question of whether the current parameterization of the stomatal control is sufficient to correctly reproducing the leaf's behavior in nature. A first approach in this regard could be the implementation of a process that artificially regulates the stomatal resistance such that the plant can stay close to its temperature optimum. Furthermore, these temperature optima can adapt to different climatological conditions (Berry and Bjorkman, 1980), which is a crucial aspect in a future warmer climate. In this sense, it would be valuable to further study the phenomenon of leaf thermoregulation, because in a warming climate

many plants may come close to their temperature limits. Thus, they would increasingly be dependent on the cooling effect due to the leaf's ability to regulate its temperature. Therefore, estimates of carbon uptake by the biosphere may crucially depend on the accurate representation of leaf thermoregulation.

*Code availability.* Access to the model source code (MPI-ESM version 2.1, ECHAM version 6.3.04, JSBACH version 3.20, gitlab.dkrz commit c2d2804b) is provided through a licensing procedure (http://www.mpimet.mpg.de/en/science/models/license/).

*Data availability.* The FLUXNET2015 dataset used for the site experiments can be found online (https://fluxnet.org/data/fluxnet2015-dataset/) and are described in detail in Pastorello et al. (2020). Information about the Tharandt site are documented in Bernhofer et al. (2016) and the data can be downloaded at https://doi.org/10.18140/FLX/1440152. For the tropical site (Goulden, 2016) the data can be accessed here: https://doi.org/10.18140/FLX/1440033. Output data, scripts used in the analysis, and other supporting information that may be useful in reproducing the authors' work can be found here ... (MPG Publications Repository (MPG.PuRe) according to copyright rules)

*Supplement.* The supplement related to this article is available online at ...

## Appendix A: Evaluation of CEBa

To test the performance of the new CEBa scheme compared to the standard JSBACH Classic, we use both FLUXNET tower sites to compare the simulated sensible and latent heat fluxes with measured eddy covariance fluxes at the tower. Therefore, we perform two single-site offline experiments with Classic and CEBa, respectively. Since the FLUXNET observations lack leaf

temperature measurements, we compare the emitted outgoing longwave fluxes instead. The radiative flux is converted into a radiative temperature $T_{\mathrm{rad}}$ using the Stefan-Boltzmann law and assuming the same emissivity of $\varepsilon = 0.996$ as used in JSBACH Classic.





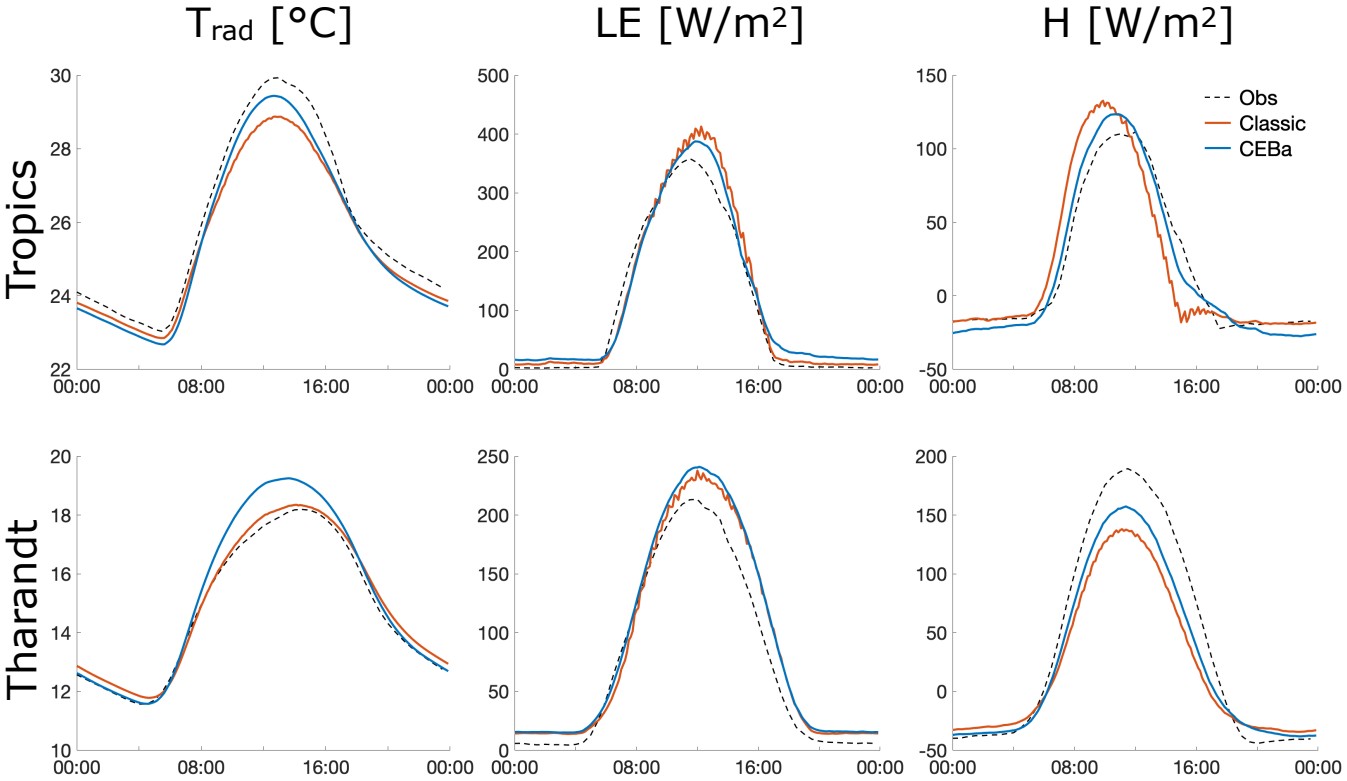

**Figure A1. Performance of the classic JSBACH scheme and JSBACH CEBa.** Displayed are comparisons of the summer season (Apr–Sep) mean diurnal cycles of the components of the canopy energy balance between JSBACH Classic (red) and JSBACH CEBa (blue) against observations (black dashed lines). Plotted are the radiative temperature $T_{rad}$ (left column), total latent heat flux $LE$ (middle column) and total sensible heat flux $H$ (right column). Data are from two FLUXNET tower sites: tropical forest in Brazil (Sept 2001 to Oct 2003, top row) and needleleaf forest in Tharandt, Germany (2006–2014, bottom row)

Figure A1 shows the comparison of the averaged diurnal cycle for the summer season (Apr–Sep) between JSBACH Classic (red) and JSBACH CEBa (blue) against observations (black dashed lines). Plotted are the radiative temperature $T_{rad}$ (left column), total latent heat flux $LE$ (middle column) and total sensible heat flux $H$ (right column). The upper row of Fig. A1 shows the comparison of fluxes for the tropical forest in Brazil for around two years from September 2001 to October 2003; the bottom row depicts the results for the needleleaf forest in Tharandt, Germany, for the years 2006 to 2014. Since the observed energy balance in Tharandt exhibits a large non-closure of almost $100\,\mathrm{W/m^2}$, the excess energy is assigned to the latent heat flux during the day and to the sensible heat flux during the night, similar to the so-called "energy residual" closure correction technique (Ershadi et al., 2014). Overall, the simulated fluxes of Classic, as well as of CEBa, fit the observations quite well. At the tropical site, Classic underestimates the daytime maximum in radiative temperature, while CEBa overestimates it for Tharandt. In general, the Classic scheme shows numerical induced fluctuations in the heat fluxes even in the average over





multiple years, while CEBa exhibits smooth curves such as seen in the observations. For the tropics, the Classic scheme shows in this regard an unusual phase shift in the sensible heat flux compared to the observations and the CEBa scheme. Since the radiative temperature is not phase-shifted, it could be caused by the abovementioned numerical instabilities rather than by incorrectly simulated heat storages. For Tharandt, both schemes show a significant difference in the Bowen ratio: the latent heat flux is overestimated, and the sensible heat flux underestimated. The cause of this bias could not be explained in this study and should be further investigated in the future.

In addition to the offline single-site experiments, we evaluated the performance of CEBa with respect to the simulated 2m-temperature in a global coupled AMIP experiment over 30 years (from 1979–2008). We find a similar pattern in the difference of the simulated near-surface temperature between CEBa and Classic as in the difference between SkIn$^+$ and Classic (see Fig. 5 in Heidkamp et al., 2018). Only a yet unexplained slight temperature increase in the high latitudes (mainly over Siberia) remains. That also leads to a significant bias reduction in the near-surface temperatures simulated by the CEBa scheme. However, despite the potentially more realistic representation of the processes in the canopy layer, the CEBa scheme does not achieve the same model bias reduction (7.4 %) as the SkIn$^+$ scheme (8.8 %).

We find that the new CEBa scheme correctly reproduces the diurnal cycle of $T_{\mathrm{sfc}}$, $H$, and $LE$ at the site-level for a temperate as well as for a tropical forest. Only for the daytime maximum in radiative temperature in Tharandt, CEBa shows a larger temperature bias than JSBACH Classic, but it removes the unrealistic drop in sensible heat storage during the afternoon at the tropical forest site. Regarding the performance of the CEBa scheme on regional and global scales, we conclude that CEBa leads to a significant bias reduction in simulated near-surface temperatures compared to JSBACH Classic, but has a slightly weaker performance than SkIn$^+$. To improve the results of the CEBa scheme, an extensive tuning and technical effort would be necessary, which is beyond the scope of this study. However, in general, the CEBa scheme provides, due to the more realistic treatment of energy exchange and its larger number of degrees of freedom, the possibility of a deeper understanding of the complex processes in the canopy layer.

## Appendix B: Logarithmic wind profile

For the estimation of $u_{\mathrm{cas}}$, we use an approach according to Xue et al. (1991). The basic idea is the concept of a so-called transition height $z_{\mathrm{trans}}$. This height can be either below or above the reference height, depending on the height of the roughness layer, which is determined by the height of the vegetation. Figure B1 illustrates the concept considering two different cases for the transition height. For tall vegetation, the first case, the transition height exceeds the reference height of the LAL $z_{\mathrm{ref}}$. For shallow vegetation, the transition height may be lower than the LAL. In this second case, the increase in wind speed of the logarithmic wind profile is scaled by an adjustment factor ($G_2 = 0.75$). Therefore, the wind speed in the CAS at the top of the vegetation $z_{\mathrm{veg}}$ can be calculated as

$$u_{\mathrm{cas}} = \begin{cases} \frac{u_* G_2}{\kappa} \ln\left(\frac{z_{\mathrm{veg}} - d}{z_0}\right) & \Rightarrow z_{\mathrm{trans}} > z_{\mathrm{ref}} \\ u_{\mathrm{ref}} - \frac{u_*}{\kappa} \ln\left(\frac{z_{\mathrm{ref}} - d}{z_{\mathrm{trans}} - d}\right) - \frac{u_* G_2}{\kappa} \ln\left(\frac{z_{\mathrm{trans}} - d}{z_{\mathrm{veg}} - d}\right) & \Rightarrow z_{\mathrm{trans}} < z_{\mathrm{ref}} \end{cases} \tag{B1}$$

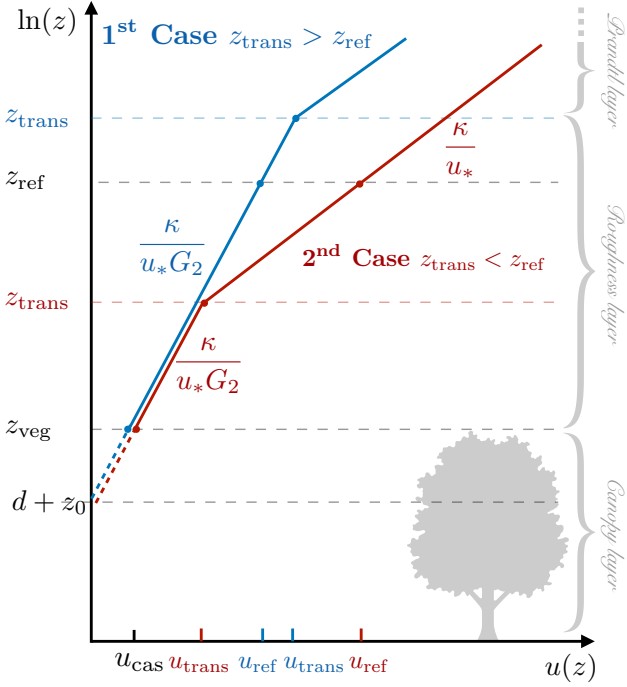

**Figure B1. Logarithmic wind profile between the zero plane displacement height and the lowest atmospheric level:** Comparison of wind profiles depending on the concept of the transition height for two different cases, where $d$ is the zero plane displacement height, $u_*$ the shear velocity, $\kappa$ the Karman constant, $G_2$ an adjustment factor, $z_{\mathrm{veg}}$ the vegetation height, $z_{\mathrm{ref}}$ the reference height, $z_{\mathrm{trans}}$ the transition height, and $u$ the wind speed at the respective height

where

$$u_* = \frac{\kappa}{G_2 \ln\left(\frac{z_{\mathrm{ref}}-d}{z_{0,\mathrm{m}}}\right)} u_{\mathrm{ref}} \tag{B2}$$

is the shear velocity and $\kappa$ the Karman constant.

**Appendix C: Physics of leaf thermoregulation in CEBa**

In the context of the CEBa scheme, the negative correlation between the leaf temperature excess and the air temperature can be explained by the energy balance of the leaf as follows: Assuming low temperatures and constant net radiation, an increase in air temperature will initially raise the leaf temperature to maintain the energy equilibrium by keeping a constant temperature difference and sensible heat flux. However, while the leaf temperature increases linearly, the saturated specific humidity of the leaf $q_{\mathrm{sat}}(T_{\mathrm{leaf}})$ increases exponentially following Clausius–Clapeyron theory. Thus, at higher temperatures the latent heat flux increasingly outweighs the sensible heat flux due to transpiration. Above a certain air temperature, the latent heat flux of the





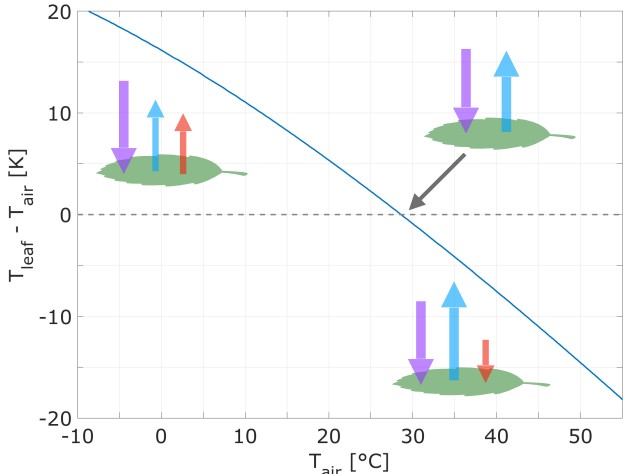

**Figure C1. Example case for the analytically solved leaf energy balance equation:** Negative correlation between leaf temperature excess and air temperature as well as three different cases of the energy balance ($R_{\text{net}}$ in violet, $H_{\text{leaf}}$ in red and $LE_{\text{leaf}}$ in blue) according to Eq. (C4).

leaf exceeds the available energy flux, resulting in a sensible heat flux towards the leaf (Fig. C1). This behavior is known as

the oasis effect and has also been found for vegetated canopies (Taha et al., 1991).

To explain this process quantitatively, we consider the energy balance of the leaf and solve it analytically for the temperature difference $T_{\text{leaf}} - T_{\text{air}}$. Ignoring heat storages within the leaf, the energy balance can be written by

$$R_{\text{net}} = LE_{\text{leaf}} + H_{\text{leaf}} \tag{C1}$$

where $R_{\text{net}}$ is the net radiation, $LE_{\text{leaf}}$ the latent- and $H_{\text{leaf}}$ the sensible heat flux of the leaf. The latent and sensible heat flux

can be expressed as

$$
\begin{aligned}
LE &= c_1 \frac{q_{\text{sat}}(T_{\text{leaf}}) - q_{\text{air}}}{r_{\text{b}} + r_{\text{s}}} \\
H &= c_2 \frac{T_{\text{leaf}} - T_{\text{air}}}{r_{\text{b}}}
\end{aligned} \tag{C2}
$$

where $q_{\text{air}}$ is the specific humidity of the air, $r_b$ is the aerodynamic leaf resistance and $r_s$ the stomatal resistance. The terms $c_1 = \rho L_{\text{v}}$ as well as $c_2 = \rho c_p$ can be seen as constant, in first order. Here $\rho$ denotes the air density, $L_{\text{v}}$ the enthalpy of vaporization and $c_p$ the specific heat capacity of air at constant pressure. The saturated specific humidity of the latent heat fluxes can

be linearized at $T = T_{\text{air}}$ as

$$LE \approx c_1 \frac{q_{\text{sat}}(T_{\text{air}}) - q_{\text{air}}}{r_{\text{b}} + r_{\text{s}}} + c_1 \alpha \frac{T_{\text{leaf}} - T_{\text{air}}}{r_{\text{b}} + r_{\text{s}}} \tag{C3}$$



where $\alpha$ is the slope of the saturation specific humidity at $T_{\mathrm{air}}$. With the abovementioned approximations, the leaf energy balance can be solved analytically and written in terms of the leaf temperature excess, as follows

$$(T_{\mathrm{leaf}} - T_{\mathrm{air}}) = \frac{r_{\mathrm{b}}}{c_2} R_{\mathrm{net}} - \frac{c_1}{c_2} \frac{1}{c_3 + r_{\mathrm{c}}/r_{\mathrm{b}}} q_{\mathrm{sat}} (T_{\mathrm{air}}) (1 - \mathrm{RH}) \tag{C4}$$

with $c_3 = 1 + \alpha c_1/c_2$. With this equation, we can quantify the atmospheric conditions and leaf properties that lead to an equilibrium between the leaf and the air temperature (equilibrium point). Equation C4 shows that an increase in net radiation or relative humidity decreases the leaf temperature excess and the plant's ability to buffer its temperature against air temperature.

*Author contributions.* All authors designed the experiments. MH performed and analyzed the simulations, developed the model code, and prepared the paper. All authors contributed to generating ideas, writing the paper, and discussing the results.

*Competing interests.* The authors declare that they have no conflict of interest.

*Acknowledgements.* The study was supported by the Max Planck Society for the advancement of science. The use of the supercomputer facilities at the Deutsches Klimarechenzentrum (DKRZ) is acknowledged. We would like to thank all people contributing to the FLUXNET network in particular these involved in the creation of the FLUXNET2015 Dataset (Pastorello et al., 2020). Especially, we thank Christian Bernhofer, Thomas Grünwald, Uta Moderow, Markus Hehn, Uwe Eichelmann, and Heiko Prasse for compiling the Tharandt dataset. The work of the tower team of the tropical FLUXNET site around Mike Goulden is also gratefully acknowledged.

The article processing charges for this open-access publication were covered by the Max Planck Society.



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
