# Peer review of "Studying the large-scale effect of leaf thermoregulation using an Earth system model"

_Earth System Dynamics, 2020_

## Referee Comment (RC1) · Anonymous Referee #1 · 30 Mar 2021

In their manuscript "Studying the large-scale effect of leaf thermoregulation using an Earth system model", the authors explore the role of leaf thermoregulation (vegetation's control on leaf temperature that can result in substantial differences between the radiative skin temperature of a leaf and the canopy air surrounding the leaf), by utilizing a canopy temperature and moisture scheme implemented into a land surface model.

The authors perform 3 distinct sets of simulations. The first solves the equations relating to canopy air space in steady-state assuming ample soil moisture. The second forces their canopy air space model with atmospheric data from FluxNet sites. The third set of simulations are coupled with an atmospheric model using MPI-ESM. They find that their simulations reproduce the expected relationship from observations, where

when air is cold, leaves are warmer than air, and when air is warm, leaves are cooler than air. They explore how this relationship (and the temperature at which leaves and air at the same T) changes for non-water-stressed vegetation based on incoming solar radiation and relative humidity. Their model predicts a lower temperature at which leaves and air are the same T compared to observations when forced with flux tower data, but this can likely largely be explained by differences in what the observations measure (top of canopy leaf temperature) and what the model simulates (temperature of vegetation across the canopy). For water-stressed forcing data and over much of the land area in the AMIP-style simulations, they find a positive (rather than negative) relationship between leaf and air temperatures. That is, while their idealized model and non-water-stressed test sites show a negative relationship, most regions in the AMIP-style simulations are actually experiencing some degree of water stress; leaf thermo-regulation should be strongest in non-water limited systems, but even the simulated leaf thermo-regulation in the AMIP simulations results in a 3-5% increase in global GPP.

The manuscript is clearly written and the study well thought out. I particularly applaud the authors on their detailed but easy-to-follow methods section. I have only minor comments on the manuscript, which are outlined point-by-point below. This manuscript is suitable for publication in Earth System Dynamics.

General comment: It might be nice to include some discussion on how much including LT modifies global evapo-transpiration and temperatures, broadly (in addition to carbon impacts)

Specific comments:

- Line 106: there are a lot of reasons that non-transpiring parts of the canopy (bark, branches/trunks) should be much hotter than transpiring leaves. Could the authors please include a discussion about if Tc here includes this and thus overestimates Tleaf, or if instead Tc really is Tleaf, and the canopy instead is estimated to all have leaf-like

temperatures? (Just make it clear if CEBa is skipping the woody-bit or of Tleaf is including the T of woody-bits in this study)

- Line 149: because the authors mention snow, what *does* happen if there is snow on the ground? Is it effectively on "top" of the vegetation, or under the vegetation but on top of the ground?

- Line 165: Do the authors mean the RH of the air space in the top soil layer? (Pardon my ignorance here, soil moisture physics are not my strongest suite). Just clarifying that they're calculating RH of air in soil vs how saturated the soil is (i.e. measuring water as a gas vs a liquid).

- Line 248-249: Might be useful here to say how much area / how much of the time the land surface isn't experiencing any water stress.

- Line 250: please give a brief explanation of the oasis effect here. (Authors do explain this near line 460, but this is the first place they mention it so it would be helpful to briefly sketch in words what it is).

- Figure 7: I assume the authors checked for all of the figure 7 cases that the negative relationship in figure 6 holds, but might be worth explicitly stating that (sorry if you did and I just missed it)

- Figure 7: clarify in legend of (d) that RH = 20% and Sin = 1000 W/m2 (and so on for other colours)

- Line 260: ie latent cooling isn't very effective when it is humid – it may be useful to reference Vargas Zepetello et al 2020. Specifically I'm thinking of figure 8 (showing changing LH has minimal effect when water flux is already large). They were more interested in soil moisture in that study, but the same general physics is at work as what I think the authors are getting at here - if you're already cooling a lot via latent cooling, and the atmosphere is really humid, it is hard to get "extra" cooling via the latent heat pathway. https://doi.org/10.1175/JCLI-D-19-0209.1

- Line 273: what do the authors mean by "vegetation period" - growing season?

---

## Referee Comment (RC2) · Anonymous Referee #2 · 12 Apr 2021

Dear Authors

This paper is potentially interesting but in the current form I have large problems and therefor suggest major revisions

Major Point 1: The authors choose to show global model performances and then only focus on two sites, tropics and Tharandt. (minor comment Tropics should be Amazon, as we don't know how it performs in other sites around the tropics)

I would prefer to have a closer look to: a) More fluxnet sites and compare the performance. Don't understand now your global model runs. If you have those, why not comparing those with all fluxnet data? At least use some other sites to see why they

deviate from each other, and why they deviate from Michaletz and other sources. Is it the LAI, type of forest, type of climate etc. I do miss a global perspective. b) To use the data available from literature, as shown in Fig.1 but also for instance by Linacre and compare those with your results. I would prefer to extend fig 1 will all your data from the introduction and make a new chapter in which you review more data available in literature. Also include in here the oxygen isotope linear regression line. It now is strange that you plot the data by Linacre but not From Helliker and others. I would prefer to have those data description in chapter 2 and then finally interesting to compare those with the model results

Major point 2: The single big leaf approach has clearly disadvantages. It is not clear to me how sunlet and shaded leaves are distinguished? Moreover, if we really want to understand the Tleaf, then we should better include the role of stomate in here (latent heat, now simply as rc?). The stomata react on T, radiation, but also on a sharp co2 gradient (higher below the canopy, specifically in the morning). I would like to see an analyses on different layers vs single layer approach and if the SBL approach still can be used to assess relialbe Tleaf, that can be verified with measurements.

Small remark L158 LAL to LAI

---

## Author Comment (AC1) · 10 May 2021

Dear Referees,

before addressing the comments in detail, we would like to thank both reviewers for taking the time to point out the shortcomings of our manuscript and to provide possible solutions to them. We value the meticulous review of our study and genuinely appreciate the efforts to address each of the issues in detail. We believe that the suggested changes significantly improve the quality of the manuscript.

**General Comment**

It might be nice to include some discussion on how much including LT modifies global evapotranspiration and temperatures, broadly (in addition to carbon impacts)

We fully agree with the reviewer and will add figures similar to Fig. 11 for temperatures (i.e. leaf and air temperature) and evapotranspiration (in the form of latent heat flux) showing the difference between the experiment where photosynthesis is calculated as a function of leaf temperature and a reference run where it is calculated as a function of air temperature (as has been the case in the past).

**Specific Comments**

- Line 106: there are a lot of reasons that non-transpiring parts of the canopy (bark, branches/trunks) should be much hotter than transpiring leaves. Could the authors please include a discussion about if Tc here includes this and thus overestimates Tleaf, or if instead Tc really is Tleaf, and the canopy instead is estimated to all have leaf-like temperatures? (Just make it clear if CEBa is skipping the woody-bit or of Tleaf is including the T of woody-bits in this study)

  Thank you for pointing this out. We added the following sentence to make it more clear:

  "This means that leaf temperature could be overestimated in the case of high solar radiation by taking into account the non-transpiring woody parts. On the other hand, the greater heat capacity of the (moist) biomass of these parts could buffer extreme temperatures and counteract this effect."

- Line 149: because the authors mention snow, what *does* happen if there is snow on the ground? Is it effectively on "topöf the vegetation, or under the vegetation but on top of the ground?

  Snow evaporation (& sublimation) is included in the model, both from the canopy but also from the ground below. Here, the treatment of snow does not differ between the standard model and our CEBa setup, and – to keep the manuscript as simple a possible – we did not include the respective terms in the equations provided in the text. To clarify this, we included the following:

"In CEBa, evaporation of snow and interception water can occur either from the top of the vegetation or from the ground below the canopy. However, for simplicity reasons, snow and interception evaporation are not included in the latent heat flux equations given below, even though they are included in the respective formulations in the model."

- Line 165: Do the authors mean the RH of the air space in the top soil layer? (Pardon my ignorance here, soil moisture physics are not my strongest suite). Just clarifying that they're calculating RH of air in soil vs how saturated the soil is (i.e. measuring water as a gas vs a liquid).

We have to agree with the reviewer that the description of relative humidity may not have been fully clear. We hope that this can be remedied by the following:

"[...],and $RH_{\mathrm{g}}$ is the relative humidity of gaseous water vapor at the soil-air interface (just above the ground), which can be parameterized by a nonlinear function depending on the water content of the top soil layer"

- Line 248-249: Might be useful here to say how much area / how much of the time the land surface isn't experiencing any water stress.

We agree that this is an important point and we would add a figure of a global map with the average (growing season) water stress in simulations with the CEBa scheme.

- Line 250: please give a brief explanation of the oasis effect here. (Authors do explain this near line 460, but this is the first place they mention it so it would be helpful to briefly sketch in words what it is).

Here, we slightly extended the sentence to provide a (very) brief summary of the oasis effect and added a reference to a paper that provides a detailed description of this effect.

"Above a certain temperature, the latent heat flux exceeds the net radiation at the surface of the leaf, resulting in the leaf absorbing energy from the surrounding air and the sensible heat flux becoming negative. This phenomenon is known as the oasis effect, which has also been found for vegetated canopies (Taha et al., 1991)."

- Figure 7: I assume the authors checked for all of the figure 7 cases that the negative relationship in figure 6 holds, but might be worth explicitly stating that (sorry if you did and I just missed it)

No, you did not miss it. One could only see it indirectly from the negative y-axis on the right side of the subfigures in Figure 7, but to make it clearer, we've also highlighted it in the text. Thank you for pointing this out.

"The slope of the regression line (SRL) holds its negative relationship (as seen in Fig. 6) for all configurations of radiation and humidity. The absolute value of the negative SRL decreases with saturation (Fig. 7b) but increases with rising radiative fluxes (Fig 7a), indicating [...]"

- Figure 7: clarify in legend of (d) that RH = 20% and Sin = 1000 W/m2 (and so on for other colours)

Thanks, we modified that in Fig. 7!

- Line 260: ie latent cooling isn't very effective when it is humid – it may be useful to reference Vargas Zepetello et al 2020. Specifically I'm thinking of figure 8 (showing changing LH has minimal effect when water flux is already large). They were more interested in soil moisture in that study, but the same general physics is at work as what I think the authors are getting at here - if you're already cooling a lot via latent cooling, and the atmosphere is really humid, it is hard to get "extra" cooling via the latent heat pathway. https://doi.org/10.1175/JCLI-D-19-0209.1

We added the reference! Thanks for the suggestion of that interesting scientific study.

- Line 273: what do the authors mean by "vegetation period growing season?

We agree that "growing season"is the better term and changed "vegetation period" to "growing season" throughout the manuscript

---

## Author Comment (AC2) · 10 May 2021

Dear Referees,

before addressing the comments in detail, we would like to thank both reviewers for taking the time to point out the shortcomings of our manuscript and to provide possible solutions to them. We value the meticulous review of our study and genuinely appreciate the efforts to address each of the issues in detail. We believe that the suggested changes significantly improve the quality of the manuscript.

**Major Point 1**

The authors choose to show global model performances and then only focus on two sites, tropics and Tharandt. (minor comment Tropics should be Amazon, as we don't know how it performs in other sites around the tropics **Thank you for pointing this out, we will change it throughout the manuscript.**) I would prefer to have a closer look to: a) More fluxnet sites and compare the performance. Don't understand now your global model runs. If you have those, why not comparing those with all fluxnet data? At least use some other sites to see why they deviate from each other, and why they deviate from Michaletz and other sources. Is it the LAI, type of forest, type of climate etc. I do miss a global perspective.)

The reviewer is correct that it may appear somewhat counter-intuitive that we spend a good deal of time discussing two site-level experiments, while the focus of the paper is the large-scale effect of leaf thermoregulation. Here, we did not include the respective sections to provide a holistic model validation, which – as the reviewer correctly pointed out – requires a comparison of numerous sites that are representative of a broad range of climate- and vegetation conditions. The site level comparison was included merely to demonstrate that the model does not reproduce the observed relation between ambient air temperature and leaf-temperature excess, even when the model is forced with the conditions that are observed at the flux-net sites, because the model estimates the average temperature of all leaves (see also response to "Major point 2") and the evaporative cooling effect is substantially reduced in a humid atmosphere and when the plants are subject to water-stress. We felt that we needed to make these points, as it is extremely counter-intuitive that on the global scale the results actually exhibit a positive correlation between ambient temperature and temperature excess (when water stress is included), while the respective formulations that were implemented in the model indicate a negative correlation (see also Fig. 6 of the original manuscript).

In general, we agree with the reviewer, that a comparison of additional sites could help provide a better picture of the model performance. However, for the following reason, we would prefer not to do this using simulations similar to the site level experiments that are already included in the manuscript: The site level simulations were not run with the standard soil/vegetation parameters (which represent a much larger area) and are also run with prescribed, observation-based, atmospheric conditions. Setting up these experiments requires a lot of time and effort, but more importantly these simulations are not necessarily representative of the standard model behaviour – i.e. when using the standard coarse resolution

parameters and when coupling JSBACH to the atmospheric model (as you can see for example in the attached figures in the appendix of this letter). Hence, we would rather propose to look at individual grid cells from the coupled (AMIP type) global run and evaluate them using site level observations – even though a good match can not necessarily be expected. Here, the required simulations have already been performed and we would make the comparison for the sites indicated in the appendix to this letter.

b) To use the data available from literature, as shown in Fig.1 but also for instance by Linacre and compare those with your results. I would prefer to extend fig 1 will all your data from the introduction and make a new chapter in which you review more data available in literature. Also include in here the oxygen isotope linear regression line. It now is strange that you plot the data by Linacre but not From Helliker and others. I would prefer to have those data description in chapter 2 and then finally interesting to compare those with the model results

In the introductory chapter, our aim was merely to provide a concise overview over the theory of leaf thermoregulation and in Fig. 1, we simply used the Linacre (1967) data as a visual support when describing the theory (please note that we added the regression lines of the in-situ measurements and the oxygen isotopes, as suggested). However – as our investigation is focused on the simulation of this effect using a large-scale model – we think that a comprehensive literature review (in a separate chapter), including a figure that combines all available observations, is beyond the scope of the present manuscript. Most importantly, the observations cannot be used for a direct comparison with our simulations, because they refer to a different scale (and predominantly to different conditions) than our model. Thus, we included the observation based regression in Fig. 8 and 9 to indicate how the effect works at the scale of individual leaves in comparison to how it works on the canopy scale. For this purpose we think that it is sufficient to show the regression line derived by Michaletz et al. (2016).

[Figure]

Fig. 1 Relation between leaf temperature excess and air temperature, based on short-term measurements of isolated and water-unstressed sunlit plants from Linacre (1967, black line) and Michaletz (2016, blue line), as well as long-term, photosynthetically weighted estimates based on cellulosic $\delta^{18}O$.

**Major point 2**

The single big leaf approach has clearly disadvantages. It is not clear to me how sunlet and shaded leaves are distinguished? Moreover, if we really want to understand the Tleaf, then we should better include the role of stomate in here (latent heat, now simply as rc?). The stomata react on T, radiation, but also on a sharp co2 gradient (higher below the canopy, specifically in the morning). I would like to see an analyses on different layers vs single layer approach and if the SBL approach still can be used to assess relialbe Tleaf, that can be verified with measurements.

We agree with the reviewer that the big leaf approach may have certain disadvantages in comparison to a multi-layer canopy scheme. However, it is computationally very efficient and consequently, used by a number of ESM land surface components, including the JS-BACH model. In the big leaf approach sunlit and shaded leaves are only implicitly separated in that the canopy has a surface area (or more precisely area that absorbs radiation) that corresponds (largely) to the sunlit leaves at the top/outside the canopy, but a heat capacity that corresponds to full canopy including the shaded leaves at the bottom/center. Thus, the reviewer is correct, that the big-leaf approach has the disadvantage that the vertical moisture/CO2/temperature structure within the canopy can not be resolved explicitly. As discussed in the manuscript, this disadvantage makes it difficult to compare the model results to observations because the model results represent the average temperatures of all sunlit and shaded leaves, while most observations pertain to exclusively to the sunlit leaves at the outside of the canopy. However, when canopy surface area and heat capacity are well parameterized – as we have spend quite a lot of work to achieve in JSBACH – the big-leaf approach is capable of capturing the overall dynamics at the coarse model resolution sufficiently well.

Unfortunately, we can not comply with the reviewer's request to compare the big-leaf approach to a multi-layer canopy scheme, as this would be a study in its own right. This would not only require implementing an entirely new canopy scheme into JSBACH but also connecting it to the rest of the ESM appropriately, e.g. implementing a vertically resolved wind profile within the canopy, which would most certainly require retuning the entire MPI-ESM. Thus, we would be happy to include a paragraph in the discussion section that addresses the shortcomings of the big-leaf approach in general, but simulations with a new canopy scheme are beyond the scope of the present study.

**Small remark**

L158 LAL to LAI

LAL here stands for *lowest atmospheric level* (as definied in line 139), not for *leaf area index*.

[Figure]

Fig. 2 See Figure 8 of the manuscript, but data derived from the global AMIP experiment (grid cell in which the Tharandt FLUXNET site is located) over 30 years (1979–2008) during growing season (Apr to Sep).

[Figure]

Fig. 3 See Figure 9 of the manuscript, but data derived from the global AMIP experiment (grid cell in which the Amazon FLUXNET site is located) over 30 years (1979–2008) during growing season (Apr to Sep).

[Figure]

Fig. 4 Global map of forest grid cells that were extracted from the global coupled AMIP run.